# Development and Control of a Real Spherical Robot

**DOI:** 10.3390/s23083895

**Published:** 2023-04-11

**Authors:** Karla Schröder, Gonzalo Garcia, Roberto Chacón, Guelis Montenegro, Alberto Marroquín, Gonzalo Farias, Sebastián Dormido-Canto, Ernesto Fabregas

**Affiliations:** 1Escuela de Ingeniería Eléctrica, Pontificia Universidad Católica de Valparaíso, Av. Brasil 2147, Valparaíso 2362804, Chile; kaivscsi@gmail.com (K.S.); garciagarreton@hotmail.com (G.G.); roberto.chacon.e@mail.pucv.cl (R.C.); jose.aguilar.m@mail.pucv.cl (A.M.);; 2Departamento de Electrotecnia e Informática, Universidad Técnica Federico Santa María, Av. Federico Santa María 6090, Viña del Mar 2520001, Chile; guelis.montenegro@usm.cl; 3Departamento de Informática y Automática, Universidad Nacional de Educación a Distancia, Juan del Rosal 16, 28040 Madrid, Spain; sebas@dia.uned.es

**Keywords:** spherical robot, electronic circuit, position control

## Abstract

This paper presents the design and implementation of a spherical robot with an internal mechanism based on a pendulum. The design is based on significant improvements made, including an electronics upgrade, to a previous robot prototype developed in our laboratory. Such modifications do not significantly impact its corresponding simulation model previously developed in CoppeliaSim, so it can be used with minor modifications. The robot is incorporated into a real test platform designed and built for this purpose. As part of the incorporation of the robot into the platform, software codes are made to detect its position and orientation, using the system SwisTrack, to control its position and speed. This implementation allows successful testing of control algorithms previously developed by the authors for other robots such as Villela, the Integral Proportional Controller, and Reinforcement Learning.

## 1. Introduction

When talking about robotics, there is a wide spectrum in terms of design, so it is very important to investigate the different types of structures that can bring a robot to life. In the case of mobile robots, they can travel through their environment using different mechanisms [1] such as wheels, legs, and caterpillars, among others, [2].

Spherical robots roll in order to move, just like a soccer ball. In particular, this type of robot possesses an inherent complexity because of its internal components. Examples of them can be found in the literature with wheel mechanisms, pendulums and omnidirectional wheels [3,4].

Nowadays, it is possible to find some types of spherical robots, which have the advantage of not being affected by overturning [5], since the interior adapts to the orientation that the robot needs to move. They are normally used for the reconnaissance of places where the terrain is not accessible [6]. On the other hand, they are used in the field of entertainment, surveillance and exploration [7]. In addition, they have a wide variety of mechanisms that allow for an effective displacement [8]. Among the uses that can be associated with this type of robot is the reconnaissance of areas in military expeditions, allowing the generation of safe spaces for soldiers [9]. Another application is extra-planetary exploration [7] since this robot is less likely to get stuck in the terrain where it moves. It is also used to explore sectors where the environment is toxic for humans and conventional robots [10]. They can also be used in the field of learning since these robots are applied to learning related to automatic control and programming. In essence, this type of robot serves as a good observer or research robot.

There are a number of mechanisms that can be used for their movement [8,11,12,13,14,15] briefly explained below. The Hamster sphere imitates the movement of the sphere with a hamster inside. It is a non-holonomic type of movement. It presents several problems in its design. The internal drive unit (IDU), the internal components of this robot are in contact with the sphere at all times. It is a low-cost robot, however at high speeds, the steering control is complex. Notable enhancements, this type of robot is known for the adaptation of the housing to the environment, in some cases, it has been modified to place a camera, a hidden leg mechanism or even to allow jumping. Multiple-Mass-Shifting, the internal mechanism is composed of three or four masses which allows changing the centre of mass to cause movement. It is of holonomic type, its position and mechanical control can become quite complex. A deformable body is based on the deformation of the casing which produces a change in the centre of mass, causing the movement. It is an incipient model, where the importance of the mechanism is in its cables since when passing current, they retract and once they cool down they return to normal. This prevents short-distance tests from being carried out. The Single ball is based on the principle of the ball mouse, considering the inverted mechanism. It requires precision in the design. Currently, there are no commercial robots of this type. Finally, there is the pendulum-driven robot, quite popular in industry and academia. The internal system contains a pendulum which swings away from the side the robot needs to go. Its speed will be determined by the weight of the pendulum. Our robot is also based on a pendulum, but it only swings forward and backward, and the orientation is achieved by rotating it. This differentiates the motion by imposing an extra difficulty in controlling the robot, derived from the frictions. To the best of our knowledge, our work represents the first time controlling an actual robot with a rotating internal pendulum, being a difference with previous works with a swinging pendulum [15,16].

This article presents the construction and testing of a real spherical robot whose motion is based on an internal pendulum. The robot consists of a spherical-shaped cover that protects the pendulum and the internal circuits that allows it to roll to move from one place to another. This type of morphology has a fundamental advantage, namely, no tipping over, which gives the robot a certain degree of stability in its movement. At the same time, it presents certain disadvantages in terms of sliding on the surface and difficulties with the presence of obstacles or irregularities in the terrain [4,17,18].

The benefits of performing tests or trials with robots built in laboratories under controlled conditions include the ability to obtain accurate results of their motions, resulting in a better comparison with simulations. Especially, since the simulations are close to ideal, and effects such as frictions of the surface or the internal mechanism, and the type of construction materials are not easily captured.

Our goal was to develop and test a simulation model of the spherical robot, as done in previous investigations [19,20] with the Khepera IV robot in the CoppeliaSim simulator [21], but implemented on a real spherical robot. For the construction, the preliminary development of a laboratory model was taken into account [22]; which is built by improvements in design and in the electronic circuit. This model was tested with various control strategies in different scenarios.

The main contributions of this article are as follows:The construction of the robot. From the 3D printing of the parts to the electronic circuit. It was optimized for greater durability in its operation.The development of the test platform. With its respective configuration to obtain the detection and orientation of the robot in a controlled space, including its code for TCP/IP communication, and for better sensing and performance (written in Python-Java).The realization of experimental tests. For this purpose, different scenarios were established for the controllers. Further, a comparative analysis with simulation results, to determine similarities or differences in all scenarios, is included. The simulator was also used for controller design.

This is a challenging task due to the complexity involved in the construction and implementation of control algorithms on a spherical robot. The developed model [23] is controlled in different scenarios with several control algorithms tested by the authors in previous studies [24], including Villela [25], IPC (proportional integral controller) [26], and reinforcement learning (RL) [27,28]. The conducted experiments for testing the mobility of the real robot including the robot design, both electronic and 3D, adjusting the position control, trajectory tracking and comparative analysis between the different scenarios. As a result of this work, the model and some examples are available online for the mobile robot community.

The article is organized as follows: Section 2 describes the construction of the spherical robot. Section 3 shows the development of the test platform. Section 4 presents the control laws used with the spherical robot. Section 5 includes a comparative analysis of the tests performed on the position control with respect to the simulation. Section 6 shows the tests performed on the position control with different targets. Finally, Section 7 presents conclusions and discusses future work.

## 2. Robot Architecture

### 2.1. 3D Design

Considering factors such as cost, the feasibility of materials, low complexity in the design, good motion control, and the possibility of construction in a short period of time, the driven pendulum mechanism was chosen as the method to be used. Therefore, the position control tests will be based on the controls carried out in article [24]. The model simulated in the article is represented by Figure 1 and considers the robot as a whole, consisting of 3 parts. The housing (a), the pendulum (b) and finally the mechanism for the electronics (c). Inspired by this simulation design, the robot is built to perform the respective tests.

For the construction of the robot, the model was designed in thinkcard 3D. In the case of the housing, 2 symmetrical pieces were printed, one red and one green. The diameter of the circumference is 20 cm and to join both pieces a thread system was devised to allow quick and easy access to the circuit.

The internal mechanism is composed of two pieces, one that holds the electronic components and the other that shapes the pendulum. The pendulum houses 0.627 kg of weight distributed evenly among its fins to generate the required moment for turning. In the centre, it has a 3.7 cm column which is connected to a DC motor. As for the inner support, this was designed to contain the servomotors, the DC motor, and the rest of the components used. It is a symmetrical piece that is long enough to reach both ends of the sphere. All parts were printed in PLA material on a Creality CR 10 max printer.

### 2.2. Electronic Circuit

Regarding the circuit, the original microcontroller ESP8266 was upgraded to an ESP32. The new microcontroller handles more memory, which allows for the management of more data, considering future works. Further, it was found that the regulator was wasting energy and increasing the temperature, so a step-down dc-dc module was added that allowed working with 5 V in the motors and feeding the ESP 32 in a more efficient way, providing more autonomy to the robot. The rest of the components used are two 360° servomotors, an XL600e step-up DC-DC converter, a DC Driver Module, a 6050 MPU, 18,560, batteries and cables. Figure 2 shows the connections of each of the components.

The robot is composed of two servomotors connected horizontally to the spherical shell allowing forward movement, and a DC motor connected to the pendulum to make the clockwise or counterclockwise movement. The three motors are driven by PWM signals to modify their speeds. For practical purposes masking tape was used to attach the components to the base, as shown in Figure 3a, where the locations of the components are specified. Finally, the Robot with its integrated parts can be seen in Figure 3b. It should be noted that for better visual detection and tracking, the original holes were covered with paper of the same colour as each semi-circumference.

## 3. Position and Orientation Detection System

The platform used in this work was designed for previous works. For example in the work with the Khepera IV robots [29], where different position control tests were performed under similar conditions. However, for this occasion, the base of the platform was substituted for a better performance of the spherical robot, i.e., reducing the friction.

Physically, the platform is composed of a USB camera fixed at the top, and a surface of 1.6×2.8 m, smooth enough to allow the robot to rotate with less friction. The platform used can be seen in Figure 4. The camera is connected to the computer, which runs the software Swistrack and the Easy Java Simulator program [30]. The router is connected to the computer, which allows communication with the robot.

The communication used was through TCP/IP between the microcontroller (Micro-Python) and the PC Module (Java). The information obtained by the USB camera is processed by Swistrack and sent to the EJS software. This in turn sorts out the data and sends it via WIFI to the robot. The Robot receives and decodes the data and then implements the position control. The computer vision-based tracking takes around 100 ms per frame to process the position and orientation sent to the robot. This delay is far greater than any delay in TCP/IP communication. On the other hand, the nature of the robot makes it a slow one, with a maximum time constant estimated in the order of 1 s. This allows for effective control despite the network and image processing delays. In Figure 5 it is possible to observe the communication between the programs and the robot.

In order for the Swistrack to send the robot’s position and orientation information, it must first be configured for sensing. To do this, we start with the calibration of the platform, indicating reference points spaced at 40 cm between the *X* and *Y* axes, [31]. Figure 6 shows what is captured by the camera. Once the above has been done, we proceed to the configuration:Input from USB Camera: Recognizes the camera to be used, in this case, it is considered −1.Timer Trigger: This trigger component processes frames (in constant intervals) at a given frame rate. Its value is 0.05 s.Convert to Color: This tool guarantees that the resulting image will be in colour (3 channels). In the case of one colour input, nothing is done. In the case of another input, the conversion is calculated.Adaptive Background: Removes the background of an image and replaces it with a black colour. Indicate Subtraction (colour), Update 0, Truncate (I-B).Red- Green Marker Detection: Detects objects that have green and red colours. Characteristics: Max 1 Blobs (Detect only 2), Max 100 pixels (of max. 100 pixels) Red Blob Threshold 55, Min blob 5 pixels, max blob 1000 pixel. Green Blob Threshold 55, Selection by area, Min blob 5 pixels, max blob 1000 pixel.Calibration with a linear Model: The calibration file of the platform must be attached, in this case, PS3-640×480.xml.Output Particles: This component writes the particle positions (and other properties).

**Figure 6 sensors-23-03895-f006:**
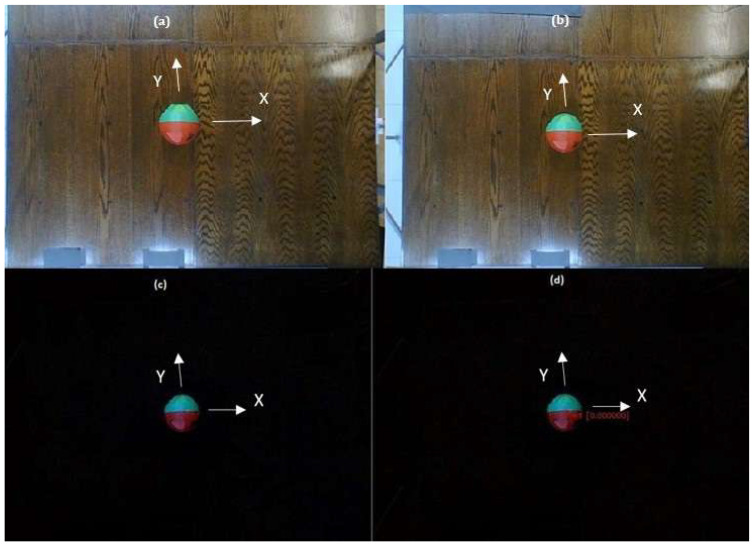
Switrack configuration: (**a**) Input from USB Camera. (**b**) Convert to colour. (**c**) Adaptive Background. (**d**) Red- Green Marker Detection.

When the robot position and orientation (xc
yc, θ) are obtained, Swistrack sends to the PC Module via TCP/IP server using the default port 3001 the data. These are written in NMEA (National Marine Electronics Association) [29], so it is necessary to decode them to be sent to the robot.

Once in the PC Module, the EJS program obtains the values received by the Swistrack and decodes them in order to obtain the values in integer format. In the case of the spherical robot, the data will be sent as a single string, including the position *X*, *Y* and the orientation θ of the robot. The latter is defined between −π to π. The Robot in turn will receive the information sent by the EJS converting the data to integers first and then floating numbers, in order to perform the position control calculations.

## 4. Position Control

Position control is basically obtained by changing the orientation of the robot as a reaction of the rotation of the internal pendulum actuated by a DC motor, and rolling the outer spherical shell as a reaction of the rotation of the servos and the effect of gravity over the pendulum. The control is then split to address lateral-directional dynamics and longitudinal dynamics, separately. The two moving mechanical pieces, the spherical shell and the pendulum are each subject to friction, so the rotation of the shell as a reaction to a constant rotation of the pendulum tends to die out in a steady state. This is evident from the dynamic model detailed in [24] where the motor torque and pendulum angular speed cancel out leaving the angular speed dynamics as a low pass filter without any input.

As shown in Figure 7 (from [24]), the idea is to control the robot to approach a target by rotating to point to the target while moving forward. This is done by reducing the angle $\alpha_{e}=\alpha -\theta$ while progressing towards the target and reducing the distance *d*. Due to the nature of this robot, the lateral-directional dynamics mainly react to a change in the pendulum’s angular speed. A proportional controller approach such as feedback of the $\alpha_{e}$ signal is not sufficient, and a new signal needs to be feedback, in this case, a slow accumulation in time of the integral of $\alpha_{e}$ called $\alpha_{e_{acc}}$. All three controllers detailed below were supplemented by the addition of this new variable in order to achieve controllability.

The control laws used in this work were designed and tested in simulation in a previous work, and were preliminary adjusted in the robot simulator software CoppeliaSim. The parameters were first adjusted by analyzing the robot’s movement in simulation. Then, due to the differences between the simulator´s model and the actual robot, the controller parameters were fine-tuned for the real experiments.

As explained, all controllers were expanded by adding the extra signal needed to overcome the nature of the robot, namely $\alpha_{e_{acc}}$. This signal is a slow accumulator version of the angle $\alpha_{e}$ given by:(1)$$\alpha_{e_{acc}}\left(t\right)=\;k_{t}\int_{t_{1}}^{t_{2}}\alpha_{e}\left(t\right)dt$$
with $k_{t}=\frac{1}{1000}$ experimentally tuned.

Common to all controllers, for lateral-directional dynamics the angle $\alpha_{e}$ and its time integral are mapped to the commanded robot’s angular speed *w*, and for the longitudinal dynamics, the distance *d* is used to command the translational speed *V*. This is shown in Figure 8.

Due to the nature of the two types of actuators, the two control signals *V* and *w* are in practice acting as virtual controls, directly setting each one the respective speeds without any significant dynamics inherently of the actuators. As is explained below these quantities are appropriately scaled and then converted onto PWM signals. The control signal *w* for Villela and IPC controllers is implemented by Equation (Equation 2).
(2)$$w\left(t\right)=-k_{\alpha_{e}}\alpha_{e}\left(t\right)-k_{\alpha_{e_{acc}}}\alpha_{e_{acc}}\left(t\right)$$

A slight modification was done to the original formulations of the controllers Villela and IPC ([24]) due to limitations in the size of the test bed. The sine function applied to $\alpha_{e}$ was not used allowing a faster correction by acting directly on the magnitude of the angle. This was significant especially at larger values, thus reducing the space required for the robot to orient itself towards the target.

### 4.1. Villela Algorithm

Considering the development of position control in article [25], it is implemented with $k_{\alpha_{e}}=2.18$, and $k_{\alpha_{e_{acc}}}=238.74$.

### 4.2. IPC Algorithm

Considering the development of position control in article [26], it is implemented with $k_{\alpha_{e}}=1.90$, and $k_{\alpha_{e_{acc}}}=238.74$.

### 4.3. Reinforcement Learning Algorithm

The control signals *w* for the Reinforcement Learning controller are implemented by:(3)$$w\left(t\right)=-k_{f_{RL}}f_{RL}\left(\alpha_{e}\left(t\right)\right)-k_{\alpha_{e_{acc}}}\alpha_{e_{acc}}\left(t\right)$$
with $k_{f_{RL}}=0.001$, and $k_{\alpha_{e_{acc}}}=31.83$, and $f_{RL}$ a function generated from the application of the reinforcement learning approach Q-learning [32,33,34]. Reinforcement learning was designed based on an explicit mathematical model detailed in the previous work [24], where the considerations for the tuning process are presented.

The generation of the control signal *V* is implemented by the following logic:(4)$$\begin{array}{c} V\left(t\right)=\left\{\begin{array}{cc} V_{max}\left(t\right) & if\left|d(t)\right|>k_{r}\\ d\left(t\right)\frac{V_{max}}{k_{r}} & if\left|d(t)\right|\leq k_{r} \end{array}\right. \end{array}$$
with $V_{max}$ around 30 cm/s and $k_{r}=10$ cm. This logic allows for a fast approach to the target while further away, and slowing it down in close proximity.

## 5. Comparative Analysis of the Real Robot v/s Simulation

Two variables are obtained from the controller, one related to the linear velocity of the robot called *V*, and the other to the angular velocity called *w*. To convert the value of *V* or *w* to a PWM signal it was necessary to perform a scaling from 0 to 1023. The movement of the robot is produced by two servo motors which move at linear speed. For this, the controller delivers a value of V, which is taken to a proportion of the PWM signal that works by letting the signal pass between 2000 and 4500 nanoseconds. Specifically, in the case of the servo, it worked a time of 2050 nanoseconds where the pulse is activated and thus the servo motor is turned on. On the other hand, the DC motor generates an angular movement in the robot, from the controller takes the value of w, which is taken to a proportion of the PWM signal that works between 20 and 100 per cent of the duty cycle (205–1023) which activates the pendulum at a certain speed. The direction of rotation is indicated by the sign of the w value. The combination of these motors allows the robot to move forward and turn to orient itself to the target point.

After performing the different tests per controller, a comparison was made with the simulation in CoppeliaSim; working with the same considerations as the article [24]. The simulator was created for preliminary controller tuning and analysis of the robot´s dynamics. The weights and physical and geometrical characteristics of the real robot were added to the simulator for good fidelity. Each of the tests was compared with the behaviour of the robot, verifying the path taken and the time took to reach the target.

### 5.1. Villela Algorithm Results

The first simulation tests were done with the Villela controller.

#### Target: $X_{p}=-80$ cm, $Y_{p}=0$ cm

In this case, the starting point of the robot is position $X_{c}=40$ cm, $Y_{c}=0$ cm and its target is at $X_{p}=-80$ cm, $Y_{p}=0$ cm. The arrow shows the initial orientation at 0 degrees. Figure 9 shows the path of the robot, as well as the time it takes to reach the destination.

It can be observed in this case that both the simulation and the behaviour of the robot, in reality, are similar in the path. However, in terms of time, the simulation takes considerably less time, reaching the target approximately 6 s earlier.

This difference can be produced by both the surface and the geometrical considerations that differ from the model built since it is based on the simulations performed in article [24] and adapted for the robot built in this article.

Finally, observing the figure it is possible to determine that the robot maintains its behaviour with respect to the path taken, in terms of time the real robot takes more time to reach the target. This can be directly related to the speed chosen for the servomotor and the difference in dynamics between the simulated and the real robot.

### 5.2. IPC Algorithm Results

The following tests were performed with the IPC controller.

#### Target: $X_{p}=-80$ cm, $Y_{p}=0$ cm

In this case, the robot’s starting point is position $X_{c}=40$ cm, $Y_{c}=0$ cm and its target is at $X_{p}=-80$ cm $Y_{p}=0$ cm. The arrow shows the initial orientation at 0 degrees. Figure 10 shows the robot’s path and the time it takes to reach the destination.

In the simulation case, the trajectory used is much broader than in the real case. Therefore, it could be said that the trajectory is not exactly the same as in the simulation. However, in terms of time, the robot maintains a similar distance-time relationship, which is held as it approaches the target. This difference in the path may be due to the same reasons stated in the Villela control.

Finally, observing the graphs, it is possible to determine that there are differences in behaviour both in the travel, since the real robot decreases its angular error faster than in the simulation, travelling less distance. This may be due to the difference between the mathematical model and reality, where both the considerations of surfaces and the geometry of the robot may not be so similar.

### 5.3. Reinforcement Learning Algorithm Results

The last controller analyzed was RL.

#### Target: $X_{p}=-80$ cm, $Y_{p}=0$ cm

In this case, the robot’s starting point is position $X_{c}=40$ cm, $Y_{c}=0$ cm and its target is at $X_{p}=-80$ cm, $Y_{p}=0$ cm. The arrow shows the initial orientation at 0 degrees. Figure 11 shows the robot’s path and the time it takes to reach the destination.

As in the first test, the behaviour of the robot is similar to the simulation, since it performs a very similar path, and in terms of the distance-time relationship, the graphs maintain a considerable similarity.

Finally, observing the graph, in this controller it can be seen that the simulation does maintain similarities compared to reality.

## 6. Experimental Results: 3 Tests for Each Controller

To check the operation of the robot driven by different controllers, tests were performed in three different scenarios which will be explained below according to the location of the target.

In the following three subsections, each figure will show the Villela controller in blue, IPC in red and Reinforcement Learning in green. They will show on the left graph the path taken by the robot to reach the endpoint, while on the right, the decrease in distance as a function of time. The same considerations when managing PWM signals apply here.

### 6.1. Target: $X_{p}=-80$ cm, $Y_{p}=40$ cm

For the first case $X_{c}=0$, $Y_{c}=-40$ was used as the starting point and $X_{p}=-80,$
$Y_{p}=40$ as the target. The arrow shows the initial orientation at 0 degrees. Figure 12 shows the results. At a glance, it can be seen that the IPC controller is the fastest with an approximate time of 9 s to reach the destination.

On the other hand, the RL controller is the one that takes the longest time with 12 s, this is due to the fact that when approaching the target it made a deviation that prevented it from arriving in less time, having to correct the orientation and therefore taking almost 3 s more.

Furthermore, the quality of each control algorithm is evaluated using performance metrics. These metrics use the error integral, which in our case is the distance to the target point. The performance measures considered in this paper are (1) total quadratic error (ISE), (2) total absolute error (IAE), (3) total time total quadratic error (ITSE) and (4) total time absolute error (ITAE).

In terms of performance indices shown in Table 1, the best values are in bold. It can be observed that the indexes are smaller in the RL control, however, when observing the graphs in Figure 12, the controller is the one that takes the longest time to reach the target.

Considering the above, it could be expected that the ITSE and ITAE indices would be higher, however, the distance that the other controllers have generates compensation. Finally obtaining a better RL index.

Although at the beginning of the path it is the first to reduce the angular error, during the path close to the target it deviates, which produces this delay in comparison with IPC, which is the most competitive after RL. In relation to Villela, it takes the longest and consequently has the worst performance indexes.

### 6.2. Target: $X_{p}=-80$ cm, $Y_{p}=0$ cm

In the second case, the starting point was $X_{c}=40$ cm, $Y_{c}=0$ cm and the target was $X_{p}=-80$ cm, $Y_{p}=0$ cm. The arrow shows the initial orientation at 0 degrees.

Figure 13 shows the results. At a glance, it can be seen that the RL control is the fastest with an approximate time of 8–9 s to reach the destination, followed by the IPC control which takes 11–12 s since it must correct its orientation along the way increasing the arrival time. While the Villela controller takes the longest time with 21–22 s since it is the longest trajectory of the 3.

In terms of performance indices shown in Table 2, the best values are in bold. It can be observed that the indices are again lower in the RL control, by observing the graphs in Figure 13, it is indeed the controller that takes the shortest time to reach the target.

On the other hand, and as in the previous case, the IPC controller is the most competitive, both in the time it takes to reach the target and in the performance index values, despite the fact that it must correct its orientation several times along the path to reach the target, unlike RL, which generates a more direct trajectory.

In relation to Villela, it takes the longest and consequently has the worst performance indexes.

### 6.3. Target: $X_{p}=-40$ cm, $Y_{p}=0$ cm

In the third case, $X_{c}=0$ cm, $Y_{c}=-40$ cm was used as the starting point and $X_{p}=-40$ cm, $Y_{p}=0$ cm as the target. The arrow shows the initial orientation at 0 degrees.

Figure 14 shows the results. At a glance, it can be seen that the RL control is the fastest as in the previous case with a time of approximately 7 s. Followed by the IPC controller, which takes approximately 9 s to reach the destination. While the Villela controller is the one that takes the longest time with 12 s, due to the distance travelled in time, as it can be seen in Figure 14.

In terms of performance indices shown in Table 3, the best values are in bold. It can be seen that the indices in the other cases are lower in the RL control. Observing the graphs in the previous figures, it can be seen that the RL controller takes the least time to reach the target, followed by the IPC control. It can also be seen from the position graph that both controllers must correct their orientation during the trajectory, which causes them to take longer to reach the target. In relation to Villela, it takes the longest and consequently has the worst performance indexes.

After analyzing the 3 cases, it can be affirmed that the RL control was the one that obtained the best yield index. In spite of the fact that on one occasion it took longer to reach the target point. The good results of the experiments can be attributed to their optimal nature given the prior training that must be performed in order to make the best approach decision to the target.

In the second place, the IPC control followed closely with very similar values. What made the difference, however, was that on two occasions, this controller took approximately one second longer to reach the target. This resulted in slightly higher rates. Both controllers had to correct their orientation along the way on several occasions during some tests, which resulted in a long time to reach the target.

In all three tests, Villela was the controller with the worst results in terms of time and performance indices. Although like the other controllers, it had to correct its orientation, it still took much longer to reach the target point.

## 7. Conclusions

This paper presents the design, construction, and implementation of a spherical robot model, whose method of motion is based on an internal pendulum. The 3D model design was developed using Tinkercad 3D modelling and design software. While the electronics were developed based on the ESP32 microcontroller, the position control strategy was programmed in Python and Java programming languages.

In order to verify its performance, different position control experiments were carried out. The results obtained with the different control laws and experiments showed that the design and implementation of the robot model were satisfactory since its behaviour was better than that obtained in the simulations.

The comparative analysis between the real robot and the simulation showed that in certain scenarios the controllers maintained similar results to the simulation, while in others, a significant difference was observed. This is mainly due to the fact that the surface friction modelled in a mathematical model is not the same as the reality, even after extensive fine-tuning. Other behavioural differences between simulation and reality are the physical/geometric structure of the CoppeliaSim which is not exactly the same as that of the robot, i.e., dimensions, mass distribution, inertias, etc.

On the other hand, the experimental tests indicated that when comparing the three controls in the simulation platform, the RL control obtains a better response, since in most of the situations it reached the target first, followed by the IPC control with very similar values.

Future work will consist of improving the communication between the PC module and the robot, which is currently done via TCP/IP, to incorporate ROS, an operating system that is much easier to use and that the microcontroller supports without problems.

## Figures and Tables

**Figure 1 sensors-23-03895-f001:**
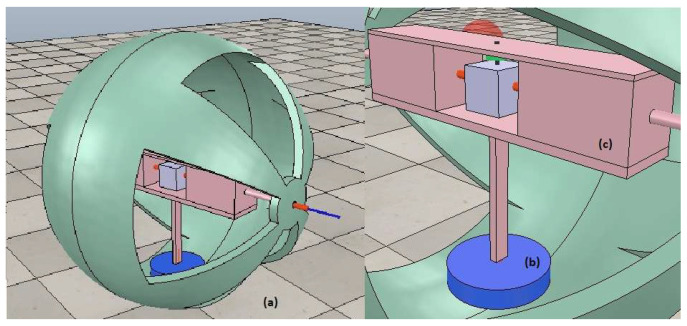
Model of the spherical robot in the simulation: (**a**) Spherical robot. (**b**) Pendulum. (**c**) Electronic mechanism.

**Figure 2 sensors-23-03895-f002:**
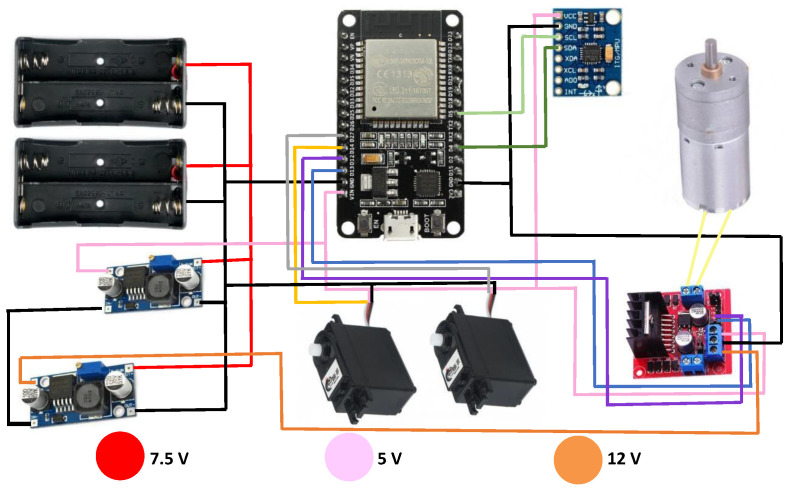
Electronic circuit used for the construction of the spherical robot.

**Figure 3 sensors-23-03895-f003:**
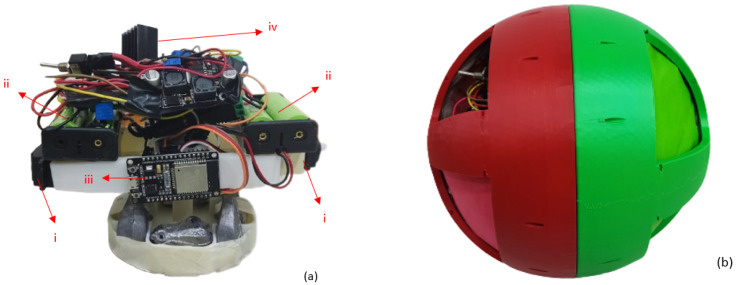
(**a**) Component locations: (**i**) 360° Servomotors. (**ii**) Batteries 18560. (**iii**) ESP32 micro-controller. (**iv**) Other components. (**b**) Spherical Robot.

**Figure 4 sensors-23-03895-f004:**
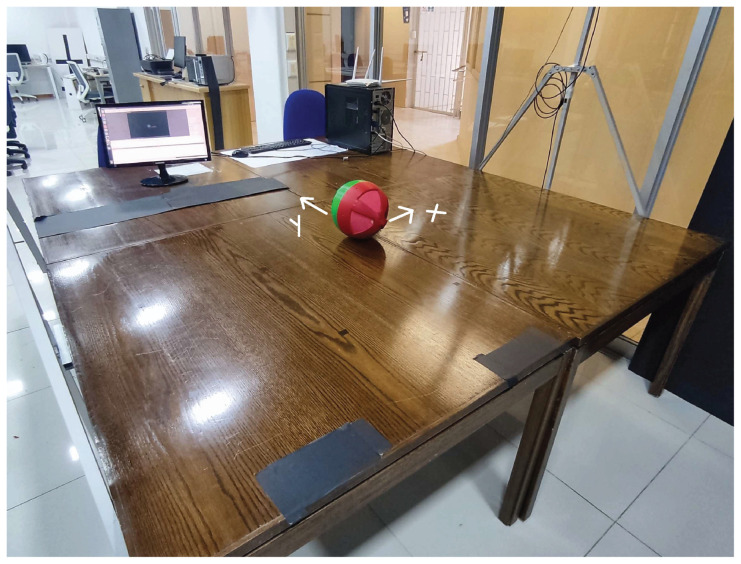
Test platform for the spherical robot.

**Figure 5 sensors-23-03895-f005:**
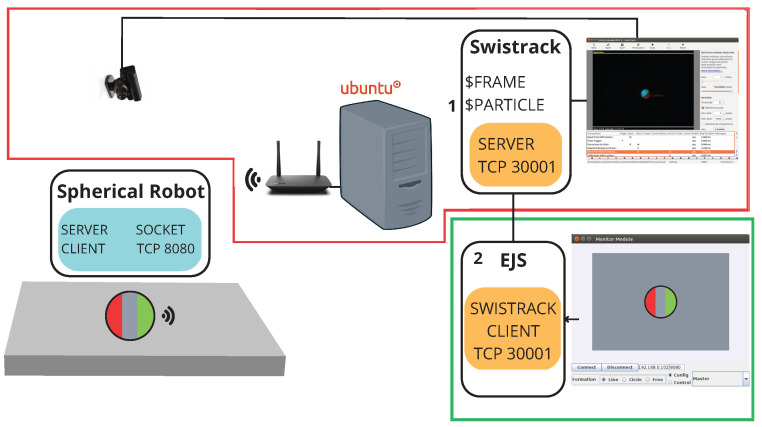
Communication PC module and spherical Robot.

**Figure 7 sensors-23-03895-f007:**
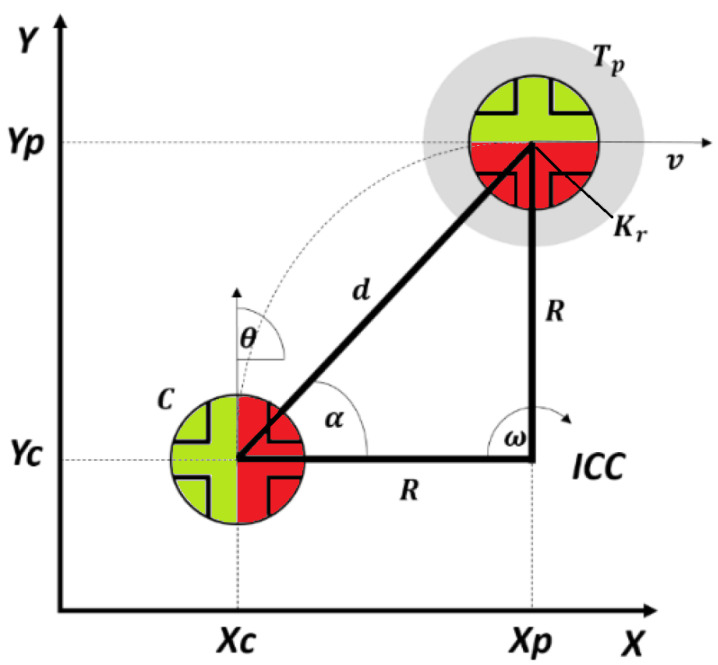
Position control geometry, from [24].

**Figure 8 sensors-23-03895-f008:**
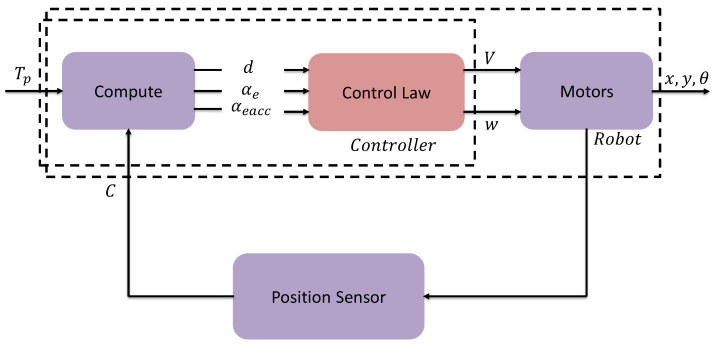
Control block diagram.

**Figure 9 sensors-23-03895-f009:**
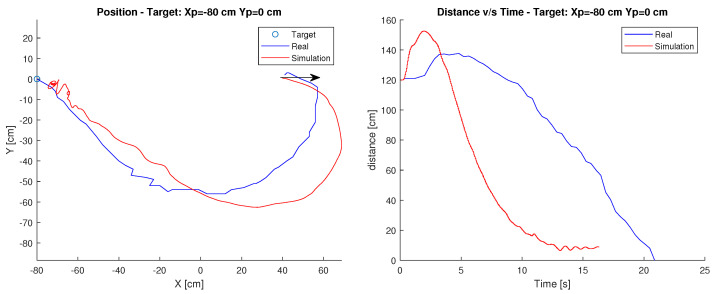
Position and distance to target $X_{p}=-80$ cm, $Y_{p}=0$ cm with the Villela controller.

**Figure 10 sensors-23-03895-f010:**
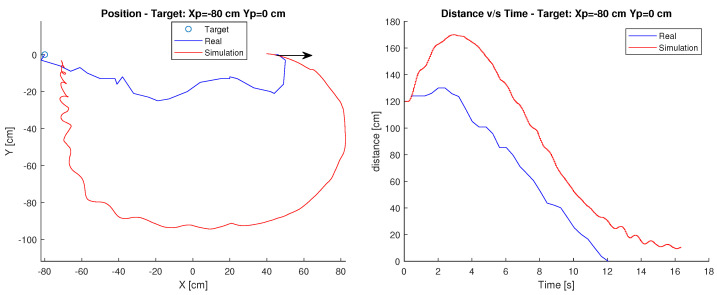
Position and distance to target $X_{p}=-80$ cm, $Y_{p}=0$ cm with the IPC controller.

**Figure 11 sensors-23-03895-f011:**
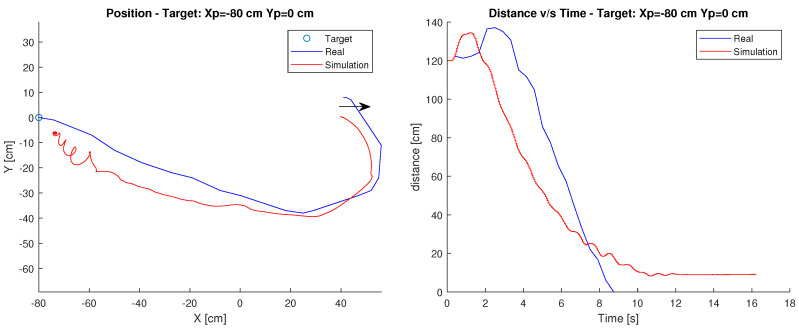
Position and distance to target $X_{p}=-80$ cm, $Y_{p}=0$ cm with the RL controller.

**Figure 12 sensors-23-03895-f012:**
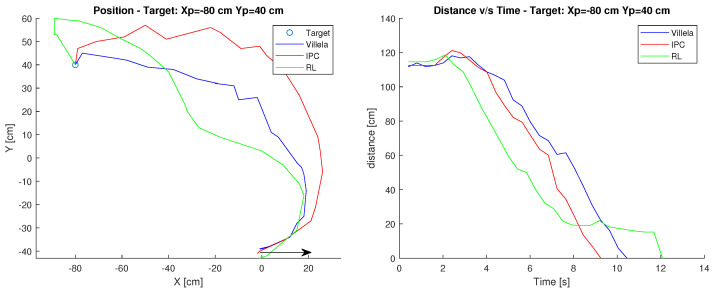
Position and distance to target $X_{p}=-80$ cm, $Y_{p}=40$ cm.

**Figure 13 sensors-23-03895-f013:**
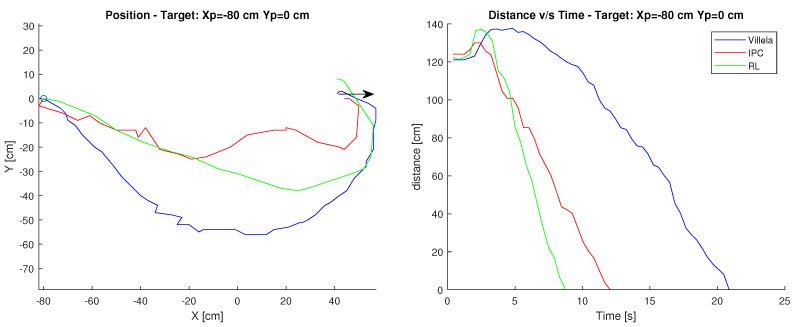
Position and distance to target $X_{p}=-80$ cm, $Y_{p}=0$ cm.

**Figure 14 sensors-23-03895-f014:**
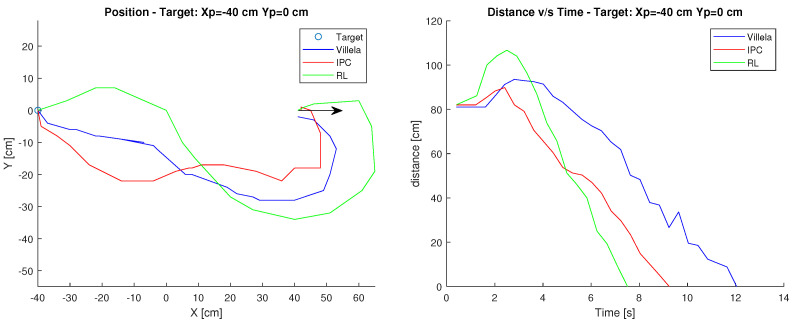
Position and distance to target $X_{p}=-40$ cm, $Y_{p}=0$ cm.

**Table 1 sensors-23-03895-t001:** Performance indexes $X_{p}=-80$ cm, $Y_{p}=40$ cm.

Indexes	Villela	IPC	RL
IAE	797	703	**641**
ISE	76,977	68,857	**53,879**
ITSE	275,800	219,640	**150,900**
ITAE	3307	2579	**2517**

**Table 2 sensors-23-03895-t002:** Performance indexes $X_{p}=-80$ cm, $Y_{p}=0$ cm.

Indexes	Villela	IPC	RL
IAE	1884	882	**710**
ISE	209,570	87,355	**77,755**
ITSE	1,470,200	311,500	**228,790**
ITAE	15,318	3878	**2408**

**Table 3 sensors-23-03895-t003:** Performance indexes $X_{p}=-40$ cm, $Y_{p}=0$ cm.

Indexes	Villela	IPC	RL
IAE	693	477	**475**
ISE	51,437	32,462	**39,758**
ITSE	208,130	96,459	**107,610**
ITAE	3248	1696	**1454**

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
