# Peer review of "Development and Control of a Real Spherical Robot"

_sensors, 2023, doi:10.3390/s23083895_

Round 1

Reviewer 1 Report

I am pleased to have read your paper.

Your research is very clear and provides sufficient background information through a thorough literature review. Your research design is reasonable and your data analysis is rigorous.

In terms of the structure of your paper, it is very clear and logical. Your data and experimental results have also been well presented and explained, enabling readers to better understand your research.

In summary, this paper can be considered for publication.

Author Response

Response to Reviewer 1 Comments

Comments and Suggestions for Authors

I am pleased to have read your paper.

Your research is very clear and provides sufficient background information through a thorough literature review. Your research design is reasonable and your data analysis is rigorous.

In terms of the structure of your paper, it is very clear and logical. Your data and experimental results have also been well presented and explained, enabling readers to better understand your research.

In summary, this paper can be considered for publication.

Authors' Response: Thank you very much for these remarks and comments.

Reviewer 2 Report

The manuscript presents the development and control for a real spherical robot. However, novelty of this manuscript is not enough for considering be published. There are some issues needed to be solved as following.

1.       The manuscript lacks convincing comparison with existed work. There are lots of paper about spherical robot, such as “Yifan Liu et al. Direction and trajectory tracking control for nonholonomic spherical robot by combining sliding mode controller and model prediction controller. 2022,https://doi.org/10.48550/arXiv.2205.14181”

2.       The robot's position control is achieved by using a DC motor to drive the internal pendulum to change the direction of the robot, and the rolling of the ball shell is driven by the servo rotation and the effect of the gravity pendulum. However, the author did not provide a detailed description of the specific implementation process of this control or the model establishment, which needs to be further elaborated.

3.       The author pointed out that the reinforcement learning algorithm used in this paper has better control effectiveness than Villela control and IPC approach. But author did not elaborate the RL algorithm related information, such as training counts or training time etc. And why RL algorithm have the best results in the experiments?

4.       In page 8, the control methods (Villela, IPC and reinforcement learning) are adopted directly from a two-wheel car control strategy (see Ref. [22]). But the dynamic model of spherical robot is different from the dynamics of the two-wheel car. The reviewer suggest author give some explanation in the paper.   

5.       It is not clear that how many Degree of freedoms and motors to actuate the robot this spherical robot have. It seems that there is only 1 DC motor for actuation, and then how does it control the lateral-directional motion and the longitudinal motion of the spherical robot?

6.  The author mentioned that the robot communicates with the upper computer through the TCP/IP communication protocol via a router, but did not indicate the potential impact of the communication delay maybe caused by this method.

Author Response

Response to Reviewer 2 Comments

The manuscript presents the development and control of a real spherical robot. However, the novelty of this manuscript is not enough for considered to be published. There are some issues needed to be solved as follows. 

  1.   The manuscript lacks a convincing comparison with existing work. There are lots of papers about spherical robots, such as “Yifan Liu et al. Direction and trajectory tracking control for the nonholonomic spherical robot by combining sliding mode controller and model prediction controller. 2022, c” 

Authors' Response: Thank you for these remarks and comments.

To the best of our knowledge, our work represents the first time controlled an actual robot with a rotating internal pendulum. The article mentioned by the reviewer together with one of its main related works, such as  “Fuzzy PID Controller Based on Yaw Angle Prediction of a Spherical Robot”, Wang, Guan, Hu, Zhang, Wang, Wang, Liu, Li, 27 September 2021, Engineering-2021 IEEE/RSJ International Conference on Intelligent Robots and Systems (IROS), are very interesting developments, but they present a robot with a non-rotating pendulum, which to our understanding, simplifies the control task. In our case, especially for rotary control, the effect of both static and dynamic friction of the pendulum and the spheric shell combined imposes a severe challenge to accurate control, where the robot tends to react to changes in the pendulum rotation. 

To clarify this point we modified the Introduction by adding the following paragraph:

“And finally, there is the pendulum-driven robot, quite popular in industry and academia. The internal system contains a pendulum which swings away from the vertical axis into the side the robot needs to go. Its speed will be mainly determined by the weight of the pendulum. Our robot is also based on a pendulum, but it only swings forward and backwards, and the orientation is achieved by rotating it. This differentiates the motion by imposing an extra difficulty in controlling the robot, derived from friction. To the best of our knowledge, our work represents the first time controlling an actual robot with a rotating internal pendulum, being a different from previous works with a swinging pendulum.

  1.       The robot's position control is achieved by using a DC motor to drive the internal pendulum to change the direction of the robot, and the rolling of the ball shell is driven by the servo rotation and the effect of the gravity pendulum. However, the author did not provide a detailed description of the specific implementation process of this control or the model establishment, which needs to be further elaborated. 

Authors' Response: Thank you for this comment, the details of the robot's motion are described in section 5, but we have included the following details.

“Two variables are obtained from the controller, one related to the linear velocity of the robot called V, and the other to the angular velocity called w. To convert the value of V or w to a PWM signal it was necessary to perform a scaling from 0 to 1023.

The movement of the robot is produced by two servo motors which move at linear speed. For this, the controller delivers a value of V, which is taken to a proportion of the PWM signal that works by letting the signal pass between 2000 and 4500 nanoseconds. Specifically, in the case of the servo, it worked a time of 2050 nanoseconds where the pulse is activated and thus the servo motor is turned on.

On the other hand, the DC motor generates an angular movement in the robot, from the controller takes the value of w, which is taken to a proportion of the PWM signal that works between 20% and 100% of the duty cycle (205 - 1023) which activates the pendulum at a certain speed. The direction of rotation is indicated by the sign of the w value.

The combination of these motors allows the robot to move forward and turn to orient itself to the target point.”

  1.       The author pointed out that the reinforcement learning algorithm used in this paper has better control effectiveness than the Villela control and IPC approach. But the author did not elaborate on the RL algorithm-related information, such as training counts or training time etc. And why RL algorithm has the best results in the experiments?

Authors' Response: Thank you for these comments.

This paper is a continuation of a previous work done by the authors (“Modeling and Control of a Spherical Robot in the CoppeliaSim Simulator”, Montenegro, Chacón, Fabregas, Garcia, Schröder, Marroquín, Dormido-Canto and Farias, Sensors 2022, 22(16), 6020), where the control algorithms were designed, tuned and tested in simulation. The present work applies the same control laws to a robot in actual experiments. The designs of the original controllers were supported by the simulator CoppeliaSim, a robot simulator which uses physics simulation libraries to perform rigid body simulation, and that allows assembling the robot graphically and setting parameters. Reinforcement learning was designed based on an explicit mathematical model detailed in the previous work, where the considerations for the tuning process are presented. The better result of RL is attributed to its optimal nature. The following paragraphs are added in sections 4 and 6 to elaborate on this:

“Reinforcement learning was designed based on an explicit mathematical model detailed in the previous work, where the considerations for the tuning process are presented.”

“After analyzing the 3 cases, it can be affirmed that the RL control was the one that obtained the best yield index. Even though on one occasion it took longer to reach the target point. The good results of the experiments can be attributed to their optimal nature given the prior training that must be performed to make the best approach decision to the target.”

  1. On page 8, the control methods (Villela, IPC and reinforcement learning) are adopted directly from a two-wheel car control strategy (see Ref. [22]). But the dynamic model of the spherical robot is different from the dynamics of the two-wheel car. The reviewer suggests the author give some explanation in the paper.    

Authors' Response: Thank you for these remarks.

It was decided to impose non-holonomic constraints for the robot to make its control more attractive from a pedagogical point of view similar to previous works: 

  • Peralta, E.; Fabregas, E.; Farias, G.; Vargas, H.; Dormido, S. Development of a Khepera IV Library for the V-REP Simulator. In Proceedings of the 11th IFAC Symposium on Advances in Control Education ACE, Bratislava, Slovakia, 1–3 June 2016; Volume 49, pp. 81–86.
  • Montenegro, G.; Chacón, R.; Fabregas, E.; Garcia, G.; Schröder, K.; Marroquín, A.; Dormido-Canto, S.; Farias, G. Modeling and Control of a Spherical Robot in the CoppeliaSim Simulator. Sensors 2022, 22, 6020.
  1.       It is not clear how many Degrees of freedom and motors to actuate the robot this spherical robot has. It seems that there is only 1 DC motor for actuation, and then how does it control the lateral-directional motion and the longitudinal motion of the spherical robot? 

Authors' Response: Thank you for these remarks and comments

The robot has two independent actuators, one DC motor oriented vertically controlling the motion of the pendulum, and two servos connected horizontally in series for the motion of the spherical shell. Both actuators are driven by PWM signals delivered by the internal electronics, which in turn are commanded by the microcontroller and its code.  The robot can reach any position and orientation in the horizontal plane, so it can be described as having 3 DOFs. At any point, it can move forward with the servos, the longitudinal motion, while steering its orientation angle with the DC motor, the rotary motion.

To improve the explanation the following changes are done in section 2:

“The robot is composed of two servomotors connected horizontally to the spheric shell allowing forward movement, and a DC motor connected to the pendulum to make the clockwise or counterclockwise movement. The three motors are driven by PWM signals to modify their speeds.”

  1. The author mentioned that the robot communicates with the upper computer through the TCP/IP communication protocol via a router, but did not indicate the potential impact of the communication delay maybe caused by this method.

             Authors' Response: Thank you for these remarks and comments

The computer vision-based tracking takes around 100 milliseconds per frame to process the position and orientation sent to the robot. This delay is far greater than any delay in TCP/IP communication. On the other hand, the nature of the robot makes it a slow one, with a maximum time constant estimated in the order of 1 second. This allows for effective control despite the network and image processing delays.

Section 3 has been modified to clarify this point.

Reviewer 3 Report

This paper presents the design and implementation of a spherical robot with an internal mechanism based on a pendulum.

I have the following remarks:

               - Paper is interesting and presents a robotic system that is a novelty in robotic architectures.

 I suggest you complete the paper with:

               - Present the dynamic model of the robot.

               - How do you determine the control law (2) of the position and orientation and how do you tune the controller parameters?

Author Response

Comments and Suggestions for Authors

This paper presents the design and implementation of a spherical robot with an internal mechanism based on a pendulum.

I have the following remarks:

- Paper is interesting and presents a robotic system that is a novelty in robotic architecture.

Authors' Response: Thank you for these remarks and comments.

 I suggest you complete the paper with:

- Present the dynamic model of the robot.

Authors' Response: Thank you for these comments.

Please note that the dynamic model was detailed in our previous work. To avoid further extending the length of the article we decided to reference it:

  • Montenegro, G.; Chacón, R.; Fabregas, E.; Garcia, G.; Schröder, K.; Marroquín, A.; Dormido-Canto, S.; Farias, G. Modeling and Control of a Spherical Robot in the CoppeliaSim Simulator. Sensors 2022, 22, 6020.

- How do you determine the control law (2) of the position and orientation and how do you tune the controller parameters?

Authors' Response: Thank you for these remarks.

The control laws used in this work were designed and tested in simulation in previous work and were preliminary adjusted in the robot simulator software CoppeliaSim.

The parameters were first adjusted by analyzing the robot's movement in simulation. Then, due to the differences between the simulator´s model and the actual robot, the controller parameters were fine-tuned for the real experiments.

Section 4 has been modified to clarify this point.

Round 2

Reviewer 2 Report

Dear author, thank you very much for the very detailed revision explanation. The reviewer considers that there is no further issue needed to clarify.